Development of a short questionnaire based on the Practice Environment Scale-Nursing Work Index in primary health care

http://orcid.org/0000-0001-8607-3195 Gea-Caballero Vicente 1 2
Juárez-Vela Raúl 3 4 raul.juarez@unirioja.es
http://orcid.org/0000-0001-6014-2180 Díaz-Herrera Miguel-Ángel 5 6
Mármol-López María-Isabel 1 2
Alfaro Blazquez Ruben 2 7
Martínez-Riera José Ramón 8
1 Nursing School La Fe, Universidad de Valencia , Valencia , Spain
2 GREIACC, Instituto de Investigación Sanitaria IIS La Fe , Valencia , Spain
3 Facultad de Ciencia y Tecnología, Universidad de La Rioja , Logroño , Spain
4 Instituto de Investigación Sanitaria de Aragón IIS-A , Logroño, Aragón , Spain
5 Primary Care Nursing Team Sant Ildefons-Cornellà 2, Institut Català de la Salut , Barcelona , Spain
6 Knowledge Mobilisation Unit, Hospital Universitari General de Catalunya , Barcelona , Spain
7 Hospital Universitari i Politècnic La Fe , Valencia , Spain
8 Departamento de Enfermería Comunitaria, Medicina Preventiva y Salud Pública e Historia de la Ciencia, Facultad de Ciencias de la Salud, Universidad de Alicante , Alicante , Spain
Simon Michael
Electronic publication date: 2019 Jul 24
Publication date: 2019
Volume: 7
Electronic Location ID: e7369
Received 2018 Dec 17; Accepted 2019 Jun 27
Copyright: © 2019 Gea-Caballero et al.
Copyright year: 2019
Copyright holder: Gea-Caballero et al.
License: This is an open access article distributed under the terms of the Creative Commons Attribution License, which permits unrestricted use, distribution, reproduction and adaptation in any medium and for any purpose provided that it is properly attributed. For attribution, the original author(s), title, publication source (PeerJ) and either DOI or URL of the article must be cited.
License URL: https://creativecommons.org/licenses/by/4.0/

Keywords: Environment, Primary health care, Community health nursing, Questionnaire design, Quality of health care

Funding: The authors received no funding for this work.

==============================
Background

Professional nursing environments determine the quality of care and patient outcomes. Assessing the quality of environments is essential to improve and obtain better health outcomes. Simplifying and shortening the way to evaluate environments reliably is also important to help nurses better understand the strengths and weaknesses of their environments. In that sense, identifying essential elements of nursing environments would allow the construction of short assessment tools to improve such environments.

Objective

To construct a short tool to assess primary health care (PHC) nursing environments based on the Practice Environment Scale-Nursing Work Index (PES-NWI) questionnaire.

Methods

Observational, cross-sectional, analytical study (data collection February–April 2015). Tool: PES-NWI (31 items). Population: PHC nurses (three health districts in Valencia, Spain) with more than 3 months in the organization. The nurses were asked to select the 10 elements of the questionnaire (items) that they considered key to facilitate and improve professional care, establishing as a final selection criterion that they obtain a global election >40%. Variables: sociodemographic and 31 questionnaire items. Analysis: descriptive statistics, reliability, multidimensional scaling (ALSCAL), factor analysis, multiple linear regression. Finally, we have analyzed the concordance between both measurements (TOP10 score on the full scale score) using the Bland–Altman method.

Results

Study sample = 269 (Response rate = 80.29%). A total of 10 elements were identified based on selection frequency of the questionnaire PES-NWI. A factorial analysis explained 62.1% of variance, internal structure of three dimensions: (1) Participation in leadership and management, (2) Nursing foundations for quality of care, (3) Adequacy of resources, with Accumulate Variance explained: Component (1): 24%; Component (2): 43.1%; Component (3): 62.1%. Reliability (Cronbach’s Alpha) was 0.816 for short questionnaire, and >0.8 for all measurements. Stress = 0.184 and RSQ = 0.793. Bland–Altman method: the scaling tends to be 1.92 points higher (equivalent to a maximum deviation of 1.54%) than the full-scale PES-NWI score (max score on PES-NWI = 124 points).

Conclusions

It is possible to identify essential elements of environments to construct a short tool that simplifies the study of PHC environments. Conducting rapid studies of environments will provide managers with information about specific elements that require prioritization to enhance quality of care and safety.

Introduction

Organization can be understood as the “process of associating or combining groups that must carry out specific envisaged actions, with the appropriate and necessary means, in order to work in a sensible, rational and coordinated manner that facilitates goal achievement” (Mompart García & Durán Escribano, 2009). Thus, nursing care does not occur in an organizational vacuum, but is the product of interaction between professionals, patients, the public, and the health service. One aspect of this interaction is the professional practice environment for nursing, which the International Council of Nurses (Baumann, 2007) has defined as “those settings that facilitate excellence and conscientious work... to ensure the health, safety and well-being of staff, promote quality patient care and improve motivation, productivity and outcomes.”

The study of nursing practice environments began with what is now considered a historic study on magnetism and health (McClure et al., 1983), and since then, significant associations have been found between optimal professional nursing practice environments and quality of care and more positive outcomes for users or patients (Copanitsanou, Fotos & Brokalaki, 2017). Excellent nursing environments yield specific benefits such as higher quality care (Kramer & Schmalenberg, 2008; Schmalenberg & Kramer, 2008; Trinkoff et al., 2010), lower rates of mortality, adverse events and work accidents (Aiken et al., 2014; Trinkoff et al., 2010), greater autonomy and professional development of clinical nurses (Kramer & Schmalenberg, 2008), lower rates of turnover, absenteeism and vacancies in the nursing team (Jones & Gates, 2007), greater staff loyalty to the organization and greater professional satisfaction (Hickson, 2013; Kelly, McHugh & Aiken, 2011), significantly lower costs and reduced administrative expenditure (Tai & Bame, 2017).

Many instruments have been developed to study and monitor nursing practice environments (Norman & Sjetne, 2017), including the Practice Environment Scale of the Nursing Work Index (PES-NWI) (A.1), developed by Lake in the USA (Lake, 2002), originally with 32 items and five dimensions. The five subscales have been shown to have an acceptable internal consistency and reliability (Cronbach’s Alpha min 0.807, and max 0.916) (Lake, 2002). This measures the characteristics of professional environments, defined as “the organisational characteristics that facilitate or constrain professional nursing practice.” The author assessed seven instruments and 54 studies of multidimensional instruments, and concluded that the PES-NWI was the most useful instrument in this respect, whilst acknowledging that none of them was brief or swift to administer. It has also been suggested that the Practice (PES-NWI) presents greater methodological strength than the other tools available (Alzate, Bayer & Squires, 2014; Gajewski et al., 2010) and is considered by most authors as the ideal instrument for assessing environments (Bonneterre et al., 2008). A more recent review also recommends its use, highlighting its “satisfactory psychometric performance, high discriminant ability, and opportunity for comparison across studies” (Swiger et al., 2017). In short, this questionnaire has contributed to the development of safe work environments and quality, efficient nursing practice (Gu & Zhang, 2014), and has been validated in various cultural and geographical contexts (Liou & Cheng, 2009; Sermeus et al., 2011). In Spain, the questionnaire was initially validated and adapted for general nursing environments with registered nurses (with one item less than the original scale) (De Pedro-Gómez et al., 2009) and later specifically for primary health care (PHC) (De Pedro-Gómez et al., 2012). Recently, it was also assessed for content validity in 33 public hospitals in the Spanish national health system (Fuentelsaz-Gallego, Moreno-Casbas & González-María, 2013). The studies conducted in Spain have mainly focused on appraising the quality of care environments in primary care. At an organizational level, primary and community care in Spain is arranged differently to hospital care. These differences are also found in the United States, where home care agencies with better-rated professional work environments offer better patient care (Jarrin, Kang & Aiken, 2017). Nurses are much more independent, manage community health, and practice within community health centers and patients’ own homes (Jarrín et al., 2014). Previous studies have shown, however, that some of the organizational characteristics present in hospital care can be equally important in community care, influencing care excellence and clinical outcomes for patients (Flynn, 2007; Jarrín et al., 2014).

In relation to the elements measured in environment assessment questionnaires, the essential elements for professional practice have been defined as “those which nurses themselves recognize as very important or significant for enhancing care in the pursuit of continuous improvement and excellence” (Kramer & Schalenberg, 2004), and various elements may be more essential than others to improve care, such as the leadership of the coordinator, interprofessional relations and the nurse’s empowerment within the organization (Anzai, Douglas & Bonner, 2014; Van Den Heede et al., 2013), even in a study about community-based settings (Jarrín et al., 2014; Mensik, 2006). The study by Mensik (2006) proposed that 10 elements were crucial for community care delivery, in agreement with other investigations conducted in hospitals (Kramer & Schalenberg, 2004). Finally, a recent experience in Spain pointed out that essential care elements could be identified by more than 40% of nurses (Gea-Caballero et al., 2017).

Despite the organizational benefits derived from the use of the tool, the author of the original questionnaire has identified the need for a short version of the PES-NWI as a priority (questionnaires evaluating environments have gradually reduced in size) (Lake, 2002), together with collecting further evidence about the questionnaire and assessing its performance in different practice environments (Lake, 2007). Having a short version of any instrument facilitates the exploration and collection of data, especially when exploring little or poorly studied environments, such as PHC, since the results in these environments have been related to burnout, satisfaction at work, quality of care and the intent to quit the job, postulated as essential information for the restructuring of work processes in the PHC environment (Lorenz & De Brito Guirardello, 2014).

Therefore, our goal was to develop a short version of the questionnaire—facilitating and simplifying data collection whilst maintaining the quality of the information obtained—by identifying the essential elements of professional nursing practice environments in PHC, that is, those elements necessary to create optimal conditions for the provision of excellent nursing care practice. A further goal was to assess the representativeness of essential items in relation to the full PES-NWI questionnaire.

Materials and Methods

Study design

Observational, cross-sectional, multicenter and analytical study conducted in 2015, in PHC units in the Xàtiva-Ontinyent, Elx-Crevillent and Torrevieja health districts (Valencia region, Spain), serving a population of 615,000 citizens.

Population and sample

The study population comprised PHC nurses working in these health districts. The questionnaire was sent to the entire population of nurses (estimating that the minimum number of responses to ensure the validity of the study was 198, with CI 95%, 5% error and a nursing population N = 335).

Inclusion and exclusion criteria

The inclusion criteria were: being a member of the health district’s permanent PHC staff, with >3 months in post. Exclusion criteria were: only nurses who did not sign the informed consent to participate were excluded from the study.

Data were not collected during the summer months (July, August, September) to avoid the rise in nurses employed temporarily to cover for those in permanent positions.

Data collection tool

We used the 31-item version of the PES-NWI questionnaire (A.p.1) validated and adapted to PHC in Spain (reliability: Cronbach’s Alpha = 0.913) (De Pedro-Gómez et al., 2012). The tool was self-completed by individuals online (Google Forms® via corporate emails). The PES-NWI encompasses five dimensions: Nurse participation in center affairs (nine items), Nursing foundation for quality of care (10 items), Management and leadership of head nurse (five items), Adequate human resources to ensure quality of care (four items), and Nurse-Physician relationship (three items). Data collection and analysis were carried out by different pairs of researchers to ensure impartiality. Researchers did not know the identity of participants.

Study variables

The sociodemographic variables collected were age (in years), gender (male, female), level of education (diploma, degree, specialist qualification, master’s degree, doctorate), professional experience (years: <2, 2–4. <4–10, >10), exercise of a management/leadership role (yes/no), health district, and place of work (Xàtiva/Ontinyent, Torrevieja, Elx/Crevillent). Each item in the questionnaire was presented as a dichotomous qualitative study variable (Nurses were asked to select the most important items to improve the care provided by them in PHC: “Yes, it is essential”/“No, it is not essential”). Which items from the PES-NWI are considered an “element” for the purposes of our study: we considered an element to be essential if it was indicated by a minimum of 40% of the study population (Gea-Caballero et al., 2017). This was also partly because previous studies in the USA had also adopted a similar top-10 approach (Mensik, 2006; Kramer & Schalenberg, 2004). The variables were grouped into the original dimensions of the PES-NWI questionnaire.

Data analysis

The statistical analysis (Alpha = 0.05) was performed with SPSS v21. In terms of descriptive statistics (%), the global reliability of the survey tool as well as all the resulting sub-scales was measured using Cronbach’s Alpha. Construct validity was measured using exploratory factor analysis with analytical validation of the degree of correlation between the variables (Kaiser-Meyer-Olkin, (KMO)) and Barlett’s test of sphericity. Factor analysis measured the total variance explained by the essential elements (“TOP10”) obtained, using principal component analysis (PCA) (Varimax-Kaiser rotation). Confirmatory analysis was conducted using multidimensional scaling (ALSCAL, with measure of S-stress and RSQ). Finally, the concordance between both measurements was analyzed using the Bland–Altman method.

Ethical aspects

Data were anonymized and protected according to relevant Spanish and European legislation (Organic Law 15/1999, European Directive 95/46/CE). The Ethics committee approved the study, and participants were provided with an information sheet and were required to sign a consent form. The authors declare no conflict of interest or funding. This research did not receive any specific grants from funding agencies in the public, commercial, or not-for-profit sectors.

Results

Descriptive results

A total of 269 nurses completed the survey (response rate 80.29%). The majority of participants were 31–40 years of age (33.1%); only 16.7% of participants were younger than 30, and 30.1% were in the 51–60 age bracket. 64.7% were women. 75.5% had more than 5 years’ experience in primary care, and 44.6% had more than 10 years’ experience. In terms of educational achievement, 79.6% nurses were university educated. Only 10.4% were managers or charge nurses.

The results are presented in Fig. 1, which identifies the 10 most essential items (TOP10) according to the ratings provided by the nurses surveyed. The cut-off at 10 items was partly determined by the nurses’ ratings, as there was a large gap between the preference for items 10 and 11; this figure was perceived to be crucial according to the participants, receiving 6.9% more selections when compared to the following element (between the last item selected from the top 10 and the eleventh item, there is a difference in the percentage of elections of 6.9%).

Figure 1 Selection (%) of each element in the PES-NWI questionnaire (numbers correspond to the item number in the original scale).

The figure shows, in % order, the frequency of the choices of each one by the nurses who participate in the study. Above 40% of elections there are a total of 10 elements of the PES-NWI questionnaire.

Analysis results

A factor analysis of the results for the full questionnaire, exploring rotated components (Varimax-Kaiser rotation), reproduced the original structure of the full questionnaire in five dimensions.

A factor analysis of the 10 essential elements, which we call the “TOP10,” explained 62.79% of the variance in three components (Accumulated Variance: Component (1): 24.96%; Component (2): 43.97%; Component (3): 62.79%).

To determine construct validity, additional exploratory factor analysis was carried out for the latent variables in the questionnaire, applying PCA. The result of the KMO test was 0.77. Bartlett’s test of sphericity was statistically significant (p < 0.001), Chi-square = 1,473.9. The results achieved in the non-parametric test to perform multidimensional scaling as alternative to the confirmatory factor analysis obtains stress values = 0.184 and RSQ coefficient = 0.793. Varimax-Kaiser rotation of the 10 essential components indicated an internal structure of three dimensions (Table 1).

Table 1 Matrix of rotateda component results (Varimax): TOP10.

Number original ítem	Essential elements (TOP10) of the PES-NWI	Components	
		1	2	3	
1	Nurses at the center have opportunities to participate in decisions that affect center policies.	0.753a	−0.098	0.195	
11	There is an active program for guaranteeing and improving quality.	0.666a	0.327	0.118	
14	The allocation of patients to each nurse promotes continuity of care (e.g., the same nurse cares for the patient over time).	−0.104	0.804a	0.183	
15	There is a common, well-defined nursing philosophy that permeates the patient care environment.	0.411	0.649a	0.207	
18	Nurses are offered continuing education programs.	0.639a	0.383	−0.058	
19	Nurses at the center present satisfactory clinical competence.	0.410	0.582a	−0.079	
20	The supervisor/coordinator is a good manager and leader.	0.685a	0.163	0.214	
25	There are sufficient employees to do the job properly.	0.104	0.082	0.939a	
26	There is a sufficient number of qualified nurses to provide quality care.	0.224	0.206	0.890a	
31	Practice is based on appropriate collaboration between nurses and physicians.	0.333	0.401a	0.197	
Note:

a Highest score.

Reliability was determined using Cronbach’s Alpha (entire questionnaire = 0.943), and for the five questionnaire dimensions (D1–D5), with all measurements obtaining >0.8 (D1 = 0.87; D2 = 0.85; D3 = 0.93; D4 = 0.84; D5 = 0.81). The reliability coefficient (Cronbach) for all of the TOP10 questionnaire items combined was 0.816. The Cronbach values for the dimensions of the short questionnaire were D1 = 0.727; D2 = 0.705; D3 = 0.899. Below, we present the TOP10 essential elements for quality care grouped into three dimensions, and define the dimensions (Table 2).

Table 2 Final structure of the scale, assigning essential items to a three-dimensional structure.

The table shows how the items are grouped in each dimension of the TOP10 questionnaire.

Dimension	Item	Item description	Normalization	
(1) Participation in management and leadership	1	Nurses at the center have opportunities to participate in decisions that affect center policies.	0.753	
11	There is an active program for guaranteeing and improving quality.	0.666	
18	Nurses are offered continuing education programs.	0.639	
20	The supervisor/coordinator is a good manager and leader.	0.685	
(2) Focus on nursing care and interdisciplinary relationships	14	The allocation of patients to each nurse promotes continuity of care (e.g., the same nurse cares for the patient over time).	0.804	
15	There is a common, well-defined nursing philosophy that permeates the patient care environment.	0.649	
19	Nurses at the center present satisfactory clinical competence.	0.582	
31	Practice is based on appropriate collaboration between nurses and physicians.	0.401	
(3) Adequate resources	25	In general, there are sufficient employees to do the job.	0.939	
26	There is a sufficient number of qualified nurses to provide quality care.	0.890	

We explored the predictive and explanatory power of the TOP10 in relation to the overall PES-NWI score in our sample (Table 3) using multiple linear regression. We found that the short scale closely predicted the overall scores obtained using the PES-NWI.

Table 3 Total variance explained by the TOP10 with respect to the original scale.

Predictive and explanatory power of the TOP10 in relation to the overall PES-NWI score in our sample, using multiple linear regression.

	% Variance explained* by the TOP10 in relation to the original five dimensions	
Overall score	90.7%	
Participation in center management	74.7%	
Focus on quality of care	86.5%	
Capacity, leadership and support of managers	55.0%	
Human resources	85.1%	
Relationships between physicians and nurses	38.0%	
Notes:

*Adjusted R2 of the multiple linear regression model.

Finally, we analyzed the concordance between both measures (PES-NWI and TOP10) using the Bland–Altman method. Previously it was found that the distribution was Gaussian and that it fulfilled all the conditions required to apply the method.

The scaling (TOP10 score on the full scale) tends to be 1.92 points higher (equivalent to a maximum deviation of 1.54%) than the full PES-NWI score (max score on PES-NWI = 124 points). The maximum differences in 95% of the cases are between −14.6 and 10.76 points. The bias of the TOP10 with respect to the PES-NWI is, therefore, 1.92 points (Fig. 2).

Figure 2 Concordance analysis between TOP10 and PES-NWI: Bland–Altman method.

The scatter plot allows to observe the high concordance between both measurements, one with the complete PES-NWI questionnaire, and with the abbreviated questionnaire TOP10.

Discussion

We aimed to synthesize and prioritize the essential elements for improving PHC, using the Spanish version of the PES-NWI questionnaire as a basis to construct a short nursing environment assessment tool. The TOP10 presents an internal structure centered around three dimensions, and the reliability—internal consistency—of the short questionnaire and its dimensions is confirmed according to Cronbach’s original criteria for short questionnaires (Cronbach, 1951). The psychometric tests performed, including Bartlett’s test of sphericity and Kaiser-Meyer, are within the intervals accepted in the literature to measure construct validity (Kaiser, 1974). When the multidimensional scaling technique was used as a non-parametric alternative to confirmatory factor analysis (Porcar Gómez & Escalante Gómez, 2009) we also obtained acceptable stress values. In addition, the Bland–Altman method has given a result that we consider good, with a high prediction power from the TOP10, in relation to the complete PES-NWI questionnaire. Overall, and based on these psychometric results, we propose a short, “TOP10” questionnaire based on the PES-NWI. If additional studies consolidate these findings, it could be a short, quick and flexible alternative option for the study and assessment of professional nursing work environments.

The acceptable percentage of variance explained by these 10 elements, together with the antecedents that already affirmed that there could be 10 key elements to explore nursing work environments (Mensik, 2006, 2007; Schmalenberg & Kramer, 2008; Gea-Caballero et al., 2017), support our focus on detecting the 10 most significant elements for nurses.

Our results are in line with those obtained by Mensik (2006, 2007) for home-care environments in the United States. Thus, our essential elements coincided with at least 10 of the elements proposed by Mensik: support from managers/administrators, focus on collaborative practices and multidisciplinary roles, partnership with physicians, interprofessional relations, promotion of professional competence, and control of contextual characteristics of the environment, which would include adequate allocation of human resources, nurse training and long-term allocation of patients to nurses (Jarrín et al., 2014; Kieft et al., 2014). With respect to their applicability in different environments, Mensik (Lake, 2002) has stated that the essential elements are probably common to or very similar in settings as diverse as hospital, community or home-based care (Mensik, 2006). Consequently, we suggest that it would be relevant and appropriate to conduct comparative research in different environments and cultures.

A study of hospital environments (Schmalenberg & Kramer, 2008) using the Essentials of Magnetism tool has indicated the essential elements of magnetism: the authors found 10 essential elements, 8 of which accounted for most of the variance and were termed the essential 8. Our findings present a high degree of agreement with these results, on up to seven items if the last item is analyzed carefully, which includes both clinical competence and training support. A recent study in Spain (Gea-Caballero et al., 2017) highlighted a number of essential elements that agree with the TOP10 proposed in the current manuscript (Table 4).

Table 4 Comparison of professional practice elements in hospital/home/community care (Kramer/Mensik/Gea 2018/Gea).

Adapted from Mensik (2006). The table shows the comparison of essential elements found in different studies. We can observe the high stability in the elements considered essential.

Organisational attribute	Staff nurses % (Kramer)	HHCa nurses % (Mensik)	% Gea-Caballero et al., (2017)	Top 10% (Gea)	
Working with other nurses who are clinically competent.	80.1	72.6	39.6	44.5b	
Good nurse/doctor relationships and communication.	79.2	60.4	43.8	56.3	
Nurse autonomy and accountability.	73.5	51.9	46.5c	(51.7)c	
Supportive nurse manager, supervisor.	69.5	80.2	48.6	60.5	
Control over nursing practice.	68.9	13.2	–	–	
Support for education.	66.2	38.7	49.3	44.5	
Adequate nursing staff.	62.5	79.2	41	47.1	
Concern for patient is paramount.	62.0	89.6	45.8	46	
Flexible work schedule.	–	67.9	–	–	
Continued competency.	–	44.3	49.3	44.5	
Adequate support services.	–	41.5	32.6	41.1	
Nurses have opportunities to participate in decisions that affect center policies.	–	–	54.2	50.6	
Notes:

a Home Health Care.

b “Working with other nurses who are clinically competent” is equalled to continued competence.

c Autonomy is not measured on the PES-NWI. Responsibility is monitored in the quality plan.

In our study, the most important factor for improving care was nursing leadership, a finding that coincides with most other studies (Jarrín et al., 2014; Mensik, 2006, 2007; Van Den Heede et al., 2013); these studies have also stressed the importance of other factors in our TOP10, for example, provision of adequate resources and good relationships between nurses/physicians.

This high level of agreement indicates that such consensus is not likely to be attributed to chance. Rather, we believe it reflects a trend in the results of the studies carried out, suggesting that, independently of the questionnaire employed or the environment studied, nurses tend to consider certain elements of particularly important to improve nursing care.

The information obtained by isolating these 10 items from the questionnaire presented a high predictive power (90.7%) in relation to the overall score obtained with the full PES-NWI questionnaire, and explained 62.79% of total variance, with a slight overestimation of 1.54% points according to the Bland–Altman method, which we consider acceptable, despite yielding broad deviation. Consequently, using our proposed TOP10 tool at an operational level (research and/or management) will yield a positive result because it provides a short, simple method to rapidly obtain reliable information on the general characteristics of a professional nursing environment. Future research is required to confirm and increase the evidence and to broaden it to the field of hospital care.

Therefore, we propose a short tool with three dimensions selected for their central role in the analysis of professional environments, and which include elements from all the dimensions in the PES-NWI; the first dimension includes items related to leadership and management of healthcare services; the second dimension relates to fundamentals of nursing for the quality of care and relations with other professionals, an aspect related to independence for decision-making and self-management of nursing practice (Burton, 2010); the third one refers to the availability of human resources. Additionally, the developer of the original PES-NWI questionnaire (Lake, 2002) considers that the item “relationships between nurses and physicians” can be confused with autonomous practice in nursing, an aspect identified by other authors (Chouinard et al., 2017). In our study, we defined that an “adequate” relationship between nurses and doctors could refer to autonomous practice and control over their sphere of practice (Kieft et al., 2014).

Construction of this short tool is in line with the recommendation of the author of the PES-NWI questionnaire (Lake, 2007), who has stressed the importance of improving evidence on the scale and constructing short versions for evaluating environments (our TOP10 proposal is administered in <2 min), as well as implementing and testing it in different nursing practice environments (PHC environments from Spain in our study, an under-researched work environment). We advocate its use in pilot evaluations of primary care environments, as well as once a complete picture of a given environment is ready, and following organizational changes in order to evaluate their impact.

We believe that short tools for assessing environments, which simplify data collection, will facilitate the evaluation and improvement of these. Consequently, the construction of a short tool based on a questionnaire such as the PES-NWI, which has been widely adapted, translated and used in many countries worldwide, is important to simplify the process of obtaining information about the most significant elements of nursing environments in order to facilitate the study and improvement of nursing work environments.

Limitations

This study is exploratory. Therefore, additional studies of practice environments with the new simplified and revised PES-NWI tool could yield further evidence concerning the validity of the TOP10 essential elements and contribute to improving quality of care by modifying these environments in order to create better conditions that make it possible to continue optimizing nursing care. It is necessary to improve the reliability of the TOP 10, as well as to reduce the deviations obtained for the short questionnaire measurements, since these are high and must, therefore, be reduced.

We are aware that our TOP10 is an unsuitable choice if the goal is to obtain exhaustive information on all five dimensions of the PES-NWI questionnaire, because it does not replicate the original structure (dimensions) and, therefore, does not have the capacity to explain the information in full. It yields equivalent information for dimensions one, two and four, but offers less information for dimensions three and five. This represents another limitation of the study, particularly with regard to D5 (Nurse-Physician relationship), which is a short dimension. However, for D3 (leadership), we believe that the element we propose is fully representative of the dimension as a whole, which could compensate for the loss of information obtained: a good leader and team coordinator ought to support the staff, see mistakes as opportunities to improve, be understanding and praise quality work.

Conclusions

Our study identified 10 key elements based on the items of the PES-NWI scale: those elements of the environment that are especially relevant to professional nursing practice in PHC. This has enabled the development of a rapid environment assessment tool consisting of 10 items (TOP10), which has shown acceptable predictive power regarding the full questionnaire.

Since professional environments and nursing activity are variable organizational factors, use of this short tool will simplify data collection and facilitate decision-making for managers in relation to improving quality and outcomes in the population.

Supplemental Information

Supplemental Information 1 Practice Environment Scale of the Nursing Work Index, Spanish Version.

Questionnaire used in the study.

Click here for additional data file.

Supplemental Information 2 Dataset used in this work.

Click here for additional data file.

Additional Information and Declarations

Competing Interests

Author Contributions

Human Ethics

Data Availability

The authors declare that they have no competing interests.

Vicente Gea-Caballero conceived and designed the experiments, performed the experiments, contributed reagents/materials/analysis tools, prepared figures and/or tables, authored or reviewed drafts of the paper, approved the final draft.

Raúl Juárez-Vela analyzed the data, prepared figures and/or tables, authored or reviewed drafts of the paper, approved the final draft, revision of the translation, adaptation of the final version.

Miguel-Ángel Díaz-Herrera performed the experiments, analyzed the data, contributed reagents/materials/analysis tools, prepared figures and/or tables, authored or reviewed drafts of the paper, approved the final draft.

Maribel Mármol López performed the experiments, contributed reagents/materials/analysis tools, prepared figures and/or tables, authored or reviewed drafts of the paper, approved the final draft, translation of the manuscript, adaptation of the final version.

Ruben Alfaro Blazquez analyzed the data, contributed reagents/materials/analysis tools, prepared figures and/or tables, authored or reviewed drafts of the paper, approved the final draft, translation of the manuscript, adaptation of the final version.

José Ramón Martínez-Riera conceived and designed the experiments, performed the experiments, authored or reviewed drafts of the paper, approved the final draft, adaptation of the final version.

The following information was supplied relating to ethical approvals (i.e., approving body and any reference numbers):

The Comité ética e investigación, Departamento de Salud Xàtiva/Ontinyent (Valencia, Spain) and the Comité ética e investigacion Departamentos de Salud Torrevieja/Elxs-Crevillent approved the study.

The following information was supplied regarding data availability:

Our raw data is available as a Supplemental File.

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
