# Peer review of "Development of a short questionnaire based on the Practice Environment Scale-Nursing Work Index in primary health care"

_PeerJ, doi:10.7717/peerj.7369_

## Round 0.1 · original submission · Major Revisions

Thanks to the reviewers we have received now two substantive reviews, which raise several points for improving the manuscript. Key points for the revision are:

1) Please provide a clear account of the decision making process for selecting these 10 items and be more clear about the to rational to derive the short version.
2) Although the English language is fairly clear it would profit from additional language editing.

Please provide a point by point response to the reviewer comments.

·

Basic reporting

Basic Reporting

The article was written in clear, unambiguous, technically correct British English. There were a few inconsistencies and errors, however. For example, the use of (,) in decimals rather than (.) and not using z in such words as organisation. There were a few words misspelled in British English, these can be corrected using search and replace features, notably psichology (psychology), analitical (analytical), dimentsions (dimensions). See the word docx version for more specific review comments and corrections.

The article included introduction and background demonstrating how the work fits with the field of nursing work environments. The author(s) especially noted previous work carried out in Spain. However, the references were somewhat dated (often 10 years old). This paper (DOI: 10.1016/j.ijnurstu.2017.06.003)
would have provided more current work in the field and supplied further support for the work.

The manuscript was organized as suggested in the Instructions for Authors.

Figures were included but in some cases they were not clearly described in the title and legends. For example, Figure 1. The title states “Selection (%) of each element in the PES-NWI questionnaire (numbers correspond to the item number in the original scale). There are many numbers in the Figure 1. The authors were likely referring to the numbering on the Y axis already labeled “ITEM”. I suggest removing the parenthetical phrase (numbers correspond to the item number in the original scale) and simply label the Y axis “Item Number on Original Scale”. Also in Figure 1, the column labeled DIFFERENCE 7.3% was confusing. I believe the difference to which this refers represents the fact that at least 7.3% of participants chose an item to be in the Top Ten over those items remaining. But in many cases the difference exceeded 7.3%. Take, for example Item 1. The difference between Items 1 and 25 (3.42% and 41.1%, respectively) is 37.68%. I suggest removing the column labeled DIFFERENCE from Figure 1.

In Tables 1 the author(s) switch from original numbering (1-31) to new numbering 1-10. Then, in Table 2 they number based on the scheme in Table 1. I found this confusing. I feel the better approach would have been to stay with the original number of items so that the reader could compare across Figure and Tables 1 and 2.

Table 3 asks a great deal of the reader who must discern which items from the original subscales are being included in the TOP TEN scale under a new latent variable structure. Perhaps a good approach, in addition to retaining the original item numbering is to include a column in Figure 1 or Table 1 showing the item’s original number and its original subscale designation. Tables are essential in making the results of a study easily discernable and authors should strive to create tables which can stand alone without requiring readers to peruse the entire text of the article in order to interpret them.

Table 4 needs some attention to detail. There is a mixture of commas and periods in decimals. Also, it would help the reader to see item number that correspond to the original 31 items. In this way figures and table can be more easily compared.

I found no reason to conclude the manuscript under review here was published elsewhere or has been subdivided inappropriately. I did find the 2015 thesis work by the primary author upon which this report was based.

See docx version for further editing comments.

Experimental design

Experimental Design (Line numbers refer to the pdf version)

The study appears to have followed prevailing ethical standards in the field and country in which the study was conducted.

The manuscript under review describes original primary research apparently drawn from an academic thesis.
It clearly defines the research question which is relevant and meaningful. The author(s) clearly show how the study builds on past work in the field and fills a gap in the need for a short survey of nurses’ work environments in primary care. The study has previously been reviewed and found rigorous by the first author’s academic review committee and having read this manuscript, I concur with limitations based on questions regarding the exact methods used.

Before reproducibility can occur the author(s) will need to provide more detailed information on methods used. Here are two examples (see the docx version for further comments):

Line 94 “…the database was refined on 2 occasions by 2 researchers to minimize error .” This statement needs further explanation. Were cases removed due to excessive missing responses or for other reasons? The author(s) reported a sample size of 268, however, when I opened the data set provided I found 269 cases. When I conducted factor analysis and chose “Listwise deletion” I had an N of 263.

Lines 154-155 I could not obtain the same alpha on the TOP TEN items as those in the manuscript. Was this due to the sample chosen and mentioned above? The number of cases I analyzed was 269

I was uncertain of the meaning of lines 168-169 What is meant by “…but lost data for dimensions 3 and 5” ?

Validity of the Findings

I had no trouble downloading and analyzing the data provided. However, until certain methods are more fully described I cannot comment on the robustness and statistical soundness of the data analyzed or any controls employed in selecting cases to analyze.

Nevertheless, taken at face value, the study appears to have followed acceptable methods of exploratory analysis and survey refinement.

Validity of the findings

Conclusions drawn (line numbers refer to the pdf version)

Lines 184-185 The author(s) state based on their results the shortened TOP TEN version of the NWI survey is “…valid, flexible, rapid and brief…” They would do well to temper their conclusions a bit more by acknowledging marginal performance of the TOP TEN version in some areas. For example:


Line 182 “…we obtained stress values < .2 (.184)….”
This reported value (.184) is considered to be fair to poor. See, for example, the generally accepted scale for Stress Values [Stress Goodness-of-fit 0.200=poor, 0.100=fair, 0.050=good, .025= excellent, 0.000=perfect] found at https://ncss-wpengine.netdna-ssl.com/wp-content/themes/ncss/pdf/Procedures/NCSS/Multidimensional_Scaling.pdf

The test for sampling adequacy (Kaiser-Meyer-Olkin) was reported to be 0.77. As in the situation mentioned above when attempting to reproduce factor results I could not reproduce the reported KMO value. Instead, I obtained 0.683 for KMO with a sample of 263. While the author(s) used a different sample size, their 0.77 is considered marginal for concluding sampling adequacy (0.8 being the usual cutoff).

Certainly, both the “marginal” results for KMO and Stress Values are quite promising. Still, I suggest that the conclusions drawn from them need a bit of tempering or qualifying.

The author points to 90% predictive power when using the TOP TEN survey to obtain the original values of NWI. This is not the most robust way to demonstrate the comparability of the two surveys. I suggest that the author(s) explore the strategy suggested by Bland and Altman Plot analysis ( https://ncss-wpengine.netdna-ssl.com/wp-content/themes/ncss/pdf/Procedures/NCSS/Bland-Altman_Plot_and_Analysis.pdf )
Bland and Altman make the point that a high correlation (or a regression) does not necessarily imply that there is good agreement between the two methods.

Even stronger would be the comparison of both versions’ sensitivity and specificiy for predicting a specified outcome. This is not necessary for this paper but should be considered as a recommendation for further study.

Additional comments

Thank you for the opportunity to review your work. I found this manuscript interesting, clear (with a few exceptions), and useful for filling gaps in the study of nursing work environments.

Your research approach was straight forward but would have been stronger if you had compared the sensitivity and specificity of the two versions of the PES-NWI (31- and 10-items) for predicting an outcome variable of interest or if you had employed the Bland Altman plot method. That is not to say that you must carry out these additional analyses for acceptance by Peer J. However, I recommend you consider these more advanced methods of evaluation in your discussion of limitations and recommendations for future research.
Of greater importance is the need for you to clarify and more fully explain the steps of your research (which I have noted in the review comments above).

·

Basic reporting

1.1 Clear and unambiguous, professional English used throughout.
- Weaknesses in English language and writing skills are quite obvious. To improve readability, formulate full sentences (for example methods part of abstract and main manuscript) and rewrite the sentences to make it more understandable and be more precise (e.g. line 78 “Through random sampling we estimated sample size to achieve representatively was 198 participants” or “Statistical analysis (alpha=.05) with SPSS v21” line 102.) Some important statements remain vague: e.g. line 47 “influencing care excellence and clinical outcomes for patients” -> what outcomes, be more specific; line 53 “various elements may be more essential than others to improve care” -> which elements? Moreover, some information seems to fit better in other sections (e.g. line 125 ”Participants were asked to select the 10 items they considered most important to help improve the care provided by nurses in primary health care.” move to method section; Discussion section should be used to discuss the results, not for repeating numbers from the results section (line 177, 179, 182). However, the main unclear aspect refers to the “elements” (what are the elements? Are they the items? Not clear how and why just 10 elements/items are selected for the short version) and the PES-NWI dimensions: instead of reducing the dimensions from 5 to 3, the authors combine two dimensions twice (Participation in management and leadership [original dimensions PES-NWI “Nurse Participation in Hospital Affairs” and “Nurse Manager Ability, Leadership, and Support of Nurses”]; Focus on nursing care and Interdisciplinary relationships [original dimensions PES-NWI “Nursing Foundations for Quality of Care” and “Collegial Nurse–Physician Relations”]. Because of this arbitrary reorganization of the original structure of the PES-NWI, the manuscript and finding remain ambiguous and rather unclear. This needs theoretical support and explanation.
- The reference style is not used consistently (e.g. line 29, 130), references are missing (e.g. line 20), for direct quotes (e.g. p. 1, line 5, 52) page numbers are missing
1.2 Abstract
The abstract is very short lacking information and not providing an adequate overview of the submitted manuscript. Use the word limit and expand the abstract with full sentences and link the parts within the text (especially method part). Moreover, including further essential information such as: Why the assessment of the nurse work environment is important? Why a short version is needed? Study sample contains what/whom? How you define an “element” (result part)?
1.3 Literature references, sufficient field background/context provided
- The first three sentences are not clear how they are related to the topic and they do not introduce very well the topic “nursing work environment”. How “organizational climate” is related to the topic and why it is in questions marks (seems it is an ambiguous concept and needs more explanation).
- Suggestion: start directly with the concept of the work environment; What is it? Why it is important? How the nurses work environment influence outcomes (Why it is important to assess the work environment)? Why PES-NWI is the “best” choice (elaborate and explain “greater methodological strength than the other tools available” (line 30). give examples and evidence underpin you statement).
- Add more latest reference for “greater staff loyalty to the organization and greater professional satisfaction” (line 21; instead of/in combination with McClure et al., 1983).
- Give information to PES-NWI (items, dimension, psychometric evaluation)
- Line 37-39: is the number of items and dimension just for the Spanish version or also for the original version from Lake? Not clear.
- What were the results from content validity analyses (line 39/40).
- “Elements” is defined with a definition (line 50.52) but what is the conclusion of this? What is an “element” in your study? Work environment dimensions? Questionnaire items? Aspects of the work environment?
- Not convincing, why exactly 10 elements are crucial and important? Why not 8 or 15?
- Line 61: Why is it needed to have a short version: “has identified the need for a short version of the PES-NWI” is also not convincing, elaborate the importance of it.
- PHC: would spell it in full because it is not common used and easily to confuse with other terms.
1.4 Professional article structure, figs, tables. Raw data shared
- Structure of the article followed is reasonable. Raw data were shared
1.5. Figures
- Descriptive results of study population would be better structured in a table to improve readability.
- Table 2: nor clear which items matched to the original scale. It seems that the wording changed. This makes it hard to compare with the original PES-NWI scale.
- Figure 1 needs further information because it is not self-explaining (how did you arrive at three groups).

Experimental design

2.1 Original primary research within Aims and Scope of the journal.
- The article is primary research and fits well in journal scope.
2.2 Research question well defined, relevant & meaningful. It is stated how research fills an identified knowledge gap.
- Research aims and study questions are not clearly formulated. Although the topic is very interesting and important for practice, the rational is not developed very well lacking to explain how the study fills this gap.
2.3 Rigorous investigation performed to a high technical & ethical standard.
- Ethical approval was given. Add a reference indicating which “Spanish and European legislation” you refer (Line 112).
- The funding statement belongs not in the method section but in a separate section at the end of the manuscript.
2.4 Methods described with sufficient detail & information to replicate.
In this section the following information is missing:
- Line 86: Is there a difference, except of the language between the original PES-NWI version from Lake and the version you used: “We used the 31-item version of the PES-NWI questionnaire (A.p.1) 87 validated and adapted to PHC in Spain”
- Line 88/89: Did the nurses have a choice whether they could complete the survey online or “in person (self administered)”
- Collect and summarize all important information of the scale in one paragraph. At the moment information are spread through the introduction and method section (within the method section in different paragraphs).
- Add information about PES-NWI: how many items in each dimension.
- Add answer options for sociodemographic variables.
- Data analyses is explained but it is not fully clear which data was included. Again, why just 10 elements/items? What if the participants would have reviewed more than ten items as “Yes, it is essential”. Seems they are lead to ten. Arguments line 127-131 are not convincing.
- Although exclusion criteria are stated, they are just formulated as inverse inclusion criteria. An exclusion criteria is an additional characteristic from the sample, that needs to be met.
- The data collection statement (line 82-84) seems to fit better in a separate paragraph “data collection”.

Validity of the findings

3.1 Data is robust, statistically sound, & controlled.
- Because the study aims and outcomes are not clear, the statistical analyses, results and conclusion seem of limited validity. Information (why a short version would make sense and which items (line 107 “essential elements “top ten”) are chosen) is not sufficiently and convincingly provided.
- Based on the combination of the two dimensions (Participation in management and leadership [original dimensions PES-NWI “Nurse Participation in Hospital Affairs” and “Nurse Manager Ability, Leadership, and Support of Nurses”]; Focus on nursing care and Interdisciplinary relationships [original dimensions PES-NWI “Nursing Foundations for Quality of Care” and “Collegial Nurse–Physician Relations”] the statement line 168/169 is rather unclear.
- Moreover, the allocation of the several items to the dimensions seems not trustworthy, e.g. item “Nurses are offered continuing education programmes” (table 2) is allocated to dimension “Participation in management and leadership” but belongs to the original PES-NWI dimensions “Nursing Foundations for Quality of Care”.
3.2 Conclusion are well stated, linked to original research question & limited to supporting results.
- Because of weaknesses in previous sections—especially introduction, method and analyses section—the discussion and conclusions are in doubt and undermines trust in the results and the connection between research and practice.

Additional comments

Thanks for submitting your manuscript ”Development of a short questionnaire based on the Practice Environment Scale-Nursing Work Index in primary health care”. It seems a very important and needed topic. However, because of several weaknesses, to my mind it does not meet expectations for publication. Please find my comments and suggestions for improvement for each section.

---

## Round 0.2 · Minor Revisions

We have now received the reviewers feedback, which is very much positive. We still found some minor revisions necessary to be addressed before acceptance of the manuscript. I've also added a pdf with some minor edits to be taken into account in the revision. Well done!

·

Basic reporting

The author(s) attended to suggestions on reporting and addressed issues pointed out by my original review. Good job!

Experimental design

Author(s) added analysis as suggested and reported that their results strengthened their conclusions.

Validity of the findings

Author(s) moderated their conclusions in light of further consideration of suggestions.

Additional comments

Your response to reviewers' comments improved your manuscript and enhances its contribution to the field of study. When you disagreed with a comment you clearly explained your rationale. Your response was clear, thoughtful, and appropriate.

·

Basic reporting

Thank you very much for re-submitting your manuscript, for incorporating much of the feedback as well as for providing justification for not making a recommended change.

The manuscript is much improved with the added information (process of establishing the TOP Ten) and the supporting analyses (Bland-Altman statistics).

Upon addressing the few remaining issues listed below, I believe that the manuscript should be considered for publication.
- Consistent use of TOP 10 or TOP Ten
- Typo in Keyword: Design
- Need to add a reference for the statement on lines 103/104 “The five subscales have been shown to have an acceptable internal consistency and reliability (Cronbach Alfa min 0.807, and max 0.916)”
- Lines 151, 158: use the abbreviation PHC
- Line 184 for variable age: add (in years)
- Line 365 D5: add in brackets the name of the subscale, as you did for D3
- Add table titles

Experimental design

no comment

Validity of the findings

no comment

---

## Round 0.3 · Minor Revisions

Well done, all edits have been satisfactory. Before we can accept your paper I would like you to further improve the figures of the manuscript. The figures will be used as you submit them. Please use the same font for both figures and without serifs (e.g. Arial).

For figure 1 - what do you mean by "Elections" please rephrase. Furthermore the plot would potentially could profit from shortened labels based on the items, the numbering is not that informative. Alternatively this could be a table.

For figure 2 - Make sure that the labeling is easy to understand.

---

## Round 0.4 · accepted · Accept

Thank you for revising your manuscript, which is now ready for publication - congratulations!